# Will COVID-19 Vaccinations End Discrimination against COVID-19 Patients in China? New Evidence on Recovered COVID-19 Patients

**DOI:** 10.3390/vaccines9050490

**Published:** 2021-05-11

**Authors:** Lu Li, Jian Wang, Anli Leng, Stephen Nicholas, Elizabeth Maitland, Rugang Liu

**Affiliations:** 1School of Business Administration, Jiangsu Vocational Institute of Commerce, Nanjing 211168, China; 200015@jvic.edu.cn; 2Dong Fureng Economic and Social Development School, Wuhan University, Beijing 100010, China; wangjian993@whu.edu.cn; 3Center for Health Economics and Management at School of Economics and Management, Wuhan University, Wuhan 430072, China; 4School of Political Science and Public Administration, Institute of Governance, Shandong University, Qingdao 266237, China; lenganli@sdu.edu.cn; 5Australian National Institute of Management and Commerce, Eveleigh, Sydney, NSW 2015, Australia; stephen.nicholas@newcastle.edu.au; 6Research Institute for International Strategies, Guangdong University of Foreign Studies, Guangzhou 510420, China; 7School of Economics, Tianjin Normal University, Tianjin 300074, China; 8School of Management, Tianjin Normal University, Tianjin 300074, China; 9Newcastle Business School, University of Newcastle, Newcastle, NSW 2308, Australia; 10School of Management, University of Liverpool, Chatham Building, Chatham Street, Liverpool L69 7ZH, UK; e.maitland@liverpool.ac.uk; 11School of Health Policy & Management, Nanjing Medical University, Nanjing 211166, China; 12Center for Global Health, Nanjing Medical University, Nanjing 211166, China

**Keywords:** COVID-19, discrimination, vaccination, determinant

## Abstract

(1) Background: By April 2021, over 160 million Chinese have been vaccinated against coronavirus disease 2019 (COVID-19). This study analyzed the impact of vaccination on discrimination against recovered COVID-19 patients and the determinants of discrimination among intended vaccinated people. (2) Methods: A self-designed questionnaire was used to collect data on COVID-19 associated discrimination from nine provinces in China. Pearson chi-square tests and a multivariate ordered logistic regression analyzed the determinants of COVID-19-related discrimination. (3) Results: People who intended to be COVID-19 vaccinated displayed a high level of discrimination against recovered COVID-19 patients, with only 37.74% of the intended vaccinated without any prejudice and 34.11% displaying severe discrimination. However, vaccinations reduced COVID-19-related discrimination against recovered COVID-19 patients from 79.76% to 62.26%. Sex, age, education level, occupation, geographical region, respondents’ awareness of vaccine effectiveness and infection risk, and COVID-19 knowledge score had a significant influence on the COVID-19 related discrimination (*p* < 0.05). (4) Conclusions: Vaccination significantly reduced COVID-19 associated discrimination, but discrimination rates remained high. Among the intended vaccinated respondents, females, the older aged, people with high school and above education level, retirees, migrant workers, and residents in central China were identified as key targets for information campaigns to reduce COVID-19 related discrimination.

## 1. Introduction

The 2019 coronavirus disease (COVID-19) pandemic impacted both the health and social well-being of populations in all countries [1]. In terms of social well-being, discrimination always follows the outbreak of infectious diseases, with the United Nations warning that “fear, rumors and stigma” would be a key challenge accompanying COVID-19 [2]. As of 1 April 2021, it has been estimated that over 100 million COVID-19 patients have recovered world-wide [3]. However, some COVID-19 recovered patients have been dismissed from their employment and some have been ostracized by their neighbors and their community [4]. Although laws have been enacted in China to protect patients with infectious diseases from discrimination, recovered COVID-19 sufferers are frequently unfairly treated in their everyday life, at work, in education and during their social interactions [5,6]. Discrimination imposes a cost not only on the individual and their families, but on the whole society through lost work and productivity [7,8]. The World Health Organization (WHO), the International Federation of Red Cross and Red Crescent Societies, and the United Nations International Children’s Emergency Fund have published guidelines to address and prevent the social stigma associated with COVID-19 [9], with the WHO Director-General calling for ‘solidarity, not stigma’ to cope with COVID-19 [10]. COVID-19 related stigma and discrimination (SAD) can cause anxiety, mental health problems and social isolation among recovered COVID-19 sufferers [11,12,13]. Such discrimination means that current COVID-19 patients may conceal their disease, fearing discrimination, which delays their treatment and imposes barriers to COVID-19 control and prevention [14,15,16]. The Chinese government is providing free COVID-19 vaccines for the entire population against COVID-19, with the National Health Commission of China reporting 160 million Chinese vaccinated by April 2021 [17]. Will COVID-19 related SAD reduce or disappear with widespread COVID-19 vaccinations? Specifically, what are the determinants of COVID-19 SAD after widespread COVID-19 vaccination?

Previous researchers have analyzed the SAD influencing factors against patients with other infectious diseases, such as hepatitis C virus (HCV), hepatitis B virus (HBV), human immunodeficiency virus (HIV) and acquired immunodeficiency syndrome (AIDS) [18,19,20,21,22,23,24]. An online anonymous nationwide survey in Japan showed that a higher level of knowledge of HBV and HCV was significantly associated with negative attitudes toward HBV and HCV-infected colleagues [24]. Yu et al. [23] found that the fear of HBV infection and lack of HBV knowledge caused HBV-related discrimination, while receiving the HBV vaccination contributed to reduce HBV discrimination in rural China. A survey of the attitudes of rural migrant workers toward HBV patients and carriers in Beijing revealed that the discrimination level was negatively associated with HBV knowledge, but positively associated with fear of HBV infection, and unemployed respondents exhibited more severe HBV-related discrimination than employed respondents [22]. Comparing HBV knowledge and stigma between chronic hepatitis B (CHB) patients and persons without HBV infection in Beijing, Huang et al. [18] found that higher stigma was associated with older age, lower education, and a lower HBV knowledge score among persons without HBV infection and that respondents with a lower education, younger age, having undergone pre-employment HBV testing regretted disclosing their CHB status. Assessing the stigma of healthy individuals toward HIV and AIDS sufferers, Jian et al [21] reported that older age adults attracted more stigma than younger adults. A survey conducted in medical providers in the southern U.S. reported that HIV knowledge had a significant effect on discrimination, prejudice, service provision and the perceived HIV risk in practice [20]. An investigation of the predictors of HIV stigma among health workers in Cape Coast, Ghana found that respondents’ opinions on HIV, and their fears and worries of becoming infected, significantly predicted their stigmatizing behavior, with nurses displaying higher stigmatizing behavior level than other health workers [19].

Most existing studies on COVID-19 SAD have been a theoretical analysis or individual opinion [25,26,27,28,29,30,31]. In a commentary, Bagcchi argued that stigma associated with COVID-19 posed a serious threat to the lives of healthcare workers, patients, and survivors of the disease, and recommended proper health education targeting the public as the most effective method to prevent social harassments of both healthcare workers and COVID-19 survivors [4]. Reporting the features of COVID-19 associated stigma in Egypt, Abdelhafiz et al., found discrimination against healthcare professionals (HCPs) and people with Asian features, calling on multi-collaboration to address COVID-19 stigma. [28]. Kyriakou et al. suggested that disseminating accurate information and providing comprehensive support to the frontline HCPs would reduce COVID-19 SAD [32]. Similarly, Logie argued that the experience of dealing with HIV could be leveraged to understand and address COVID-19 stigma [30]. In another study, Logie and Turan suggest that balancing measures of containment and prevention of the pandemic could help reduce the COVID-19 stigma [27]. Based on news published online or in print in India, Bhanot et al. argued stigma reduced the health and treatment-seeking behavior of COVID-19 sufferers, suggesting all relevant stakeholders ought to mitigate stigma through a multipronged approach [26]. Using case studies, Grover et al. recommended the dissemination of information about the mode of transmission and the importance of testing to address discrimination against HCPs [29]. For COVID-19, Daalen et al. [25] recommended timely and honest risk communication, addressing misinformation and improving awareness; ensuring employment sick leave and access to testing; and implementing of skill training and educational programs.

There has been no quantitative study of SAD against recovered COVID-19 patients in China, and little is currently known about the determinants of COVID-19 related SAD by the COVID-19 vaccinated. To address this lacuna, our study assessed the impact of COVID-19 vaccinations on reducing discrimination against recovered COVID-19 patients; analyzed the determinants of the discrimination against COVID-19 recovered patients when people intended to be vaccinated against COVID-19; and recommends measures to reduce COVID-19 related discrimination when the population has received the COVID-19 vaccines. The main hypothesis of this study is that vaccination should reduce COVID-19 related discrimination. Second, the study assesses the potential determinants of discrimination even after people have received the COVID-19 vaccine.

## 2. Materials and Methods

### 2.1. Data Source and Sample

A questionnaire was designed to collect nationwide data on discrimination against recovered COVID-19 patients and respondents’ socioeconomic and demographic variables and knowledge of COVID-19 and COVID-19 vaccines. China’s 27 provinces were divided into three regions: eastern, central, and western. The provinces in each region were stratified into low, medium, and high economic levels according to their 2019 gross domestic product (GDP). Randomly, one province was chosen from each economic level in each region, yielding nine provinces. Next, according to their 2019 GDP rank, all the cities in each selected province were divided into low, medium, and high economic levels. One city was randomly chosen from each GDP city-level from each province, with 27 cities selected from the 9 provinces. In each city, one hundred participants were interviewed face-to-face, or by online video interviews in cities where participants were required to home quarantine, with equal numbers of men and women and three urban residents for every two rural residents, which reflected the nationwide urban–rural breakdown. All investigators recruited in the 27 cities received standardized training before the interviews. Before the roll-out of the nationwide vaccination program in China, face-to-face interviews were conducted from 30 May to 10 June 2020 on respondents who intended to COVID-19 vaccinate and the unvaccinated [33]. All participants were informed about the purpose of the survey and gave informed consent. We collected data on 2700 adults over the age of 18 years old, which yielded a sample of 2377 respondents after deleting cases with missing data, with a response rate of 88.04%.

### 2.2. Definition and Measurement of Discrimination against Recovered COVID-19 Patients

As shown in Table 1, the discrimination against recovered COVID-19 patients was measured by asking respondents who intended to COVID-19 vaccinate and unvaccinated respondents about their attitudes to six life events. To each question in Table 1, respondents answered “yes” (coded 0), “it depends” (coded 1) and “no” (coded 2) representing low, medium, and high COVID-19 related discrimination. The sum of their response scores across the six events yielded a discrimination index, which ranged from 0 to 12, where higher scores indicated greater discrimination. Participants were then categorized into three discrimination levels, no discrimination (score 0), medium discrimination (scores 1–5), and severe discrimination (scores 6–12).

### 2.3. Definition and Measurement of Independent Variables

As shown in Table 1, the independent variables comprised sex (male–female), age groups, five average monthly income groups (very low < RMB 2500; low ≥ RMB 2500–< RMB4000; medium ≥ RMB4000–< RMB6000; high (≥RMB6000–< RMB 10000; and very high ≥ RMB 10000), education level (primary school and below, middle school, high school and above high school), occupation, medical insurance (yes/no), urban–rural residence, self-rated health, residence in east–west–central regions, and awareness of COVID-19 vaccine effectiveness andCOVID-19 infection risk, experience of paying for vaccines, whether their relatives or friends has been infected by COVID-19 and knowledge score about COVID-19. Reflecting the national urban–rural population distribution, three urban residents were interviewed for every two rural participants. Occupations were categorized into physicians, farmers, migrant workers, employees, self-employed, teacher, civil servant, professional and technical staff, unemployed, student, retiree, and other occupation. Self-rated health was categorized into, “bad”, “medium” and “good”, based on the question: “How is your health status compared to your peers?” Participant’s awareness of COVID-19 vaccine effectiveness was measured by the questions: “Do you believe that the COVID-19 vaccine is effective?”, and coded into a three-item Linkert scale (“don’t agree—low effectiveness”, “neutral attitude—medium effectiveness” and “agree—high effectiveness”). The respondents’ risk of infection was measured by asking participants whether they felt they will be infected by COVID-19 in the future according to “low–neutral–high” measure of risk. Whether participants had paid for non-COVID-19 vaccines for their family members or themselves in the past was scored “never”; “paid in the last year”; and “paid more than one year ago”. Participants were also asked whether their relatives or friends were infected by COVID-19, selecting “no”, “yes” and “not sure”.

Respondents’ COVID-19 knowledge were measured by positive and negative points on correct and incorrect answers to five multiple-choices question on transmission (with two true and two false answers); prevention knowledge (with three true and two false answers); susceptible population; the number of isolation days; and therapeutic method. The sum score of the five knowledge questions ranged from −4 to 8. The respondents then were divided into low (≤5) and high (>5) COVID-19 knowledge score groups by the median of the sum knowledge score.

### 2.4. Statistical Analyses

All data were double-entered using EpiData 3.1 and checked for consistency. Statistical analyses were performed using STATA 12.0. The Pearson chi-square test was used to compare the differences of discrimination level between unvaccinated and intended vaccinated respondents. Multivariate ordered logistic regression models were used to assess the associations between each independent variable and the COVID-19 associated discrimination among vaccinated respondents.

## 3. Results

### 3.1. Characteristics of Respondents

Table 2 shows the characteristics of 2377 survey respondents, with the sex ratio broadly even and the urban –rural split close to the 3:2 national urban–rural ratio; the median age was 35; the median monthly income was RMB5000 and the five income groups were roughly equal. In terms of education, 56.84% respondents had an above high school education level. Students (26.88%) accounted for the highest occupational group, followed by employees (13.59%), migrant workers (12.16%) and farmers (11.70%). Only 3.07% reported their self-assessed health as “bad” and 96% had medical insurance. There was a broadly equal number of respondents from the eastern, central, and western regions. Respondents mostly believed that the COVID-19 vaccine was effective (86.62%); 53.43% thought they were at medium or high risk of COVID-19 infection; and 56.75% of respondents reported they had previously paid for vaccines for their family members or themselves. Only 1.51% had relatives or friends who had been infected by COVID-19. The median of COVID-19 knowledge score was 5.

### 3.2. Discrimination against COVID-19 Patients among Subgroups of Respondents

Pearson chi-square tests in Table 2 show that there were significant correlations between the level of discrimination and sex, age, education, occupation, region, vaccine effectiveness, risk of infection, experience with paid vaccines and COVID-19 knowledge score (*p* < 0.05).

Females’ severe discrimination level (35.9%) was significantly higher than males’ (32.24%) (*p* = 0.001); the percent of severe discrimination rose with age; severe discrimination increased as respondents’ cognition of infection risk increased, but severe discrimination rapidly decreased as cognition of COVID-19 vaccine effectiveness rose (*p* < 0.01). People aged over 58 years old (48.73%), believing the COVID-19 vaccine efficacy was low (76.47%) and feeling they were in high risk of COVID-19 infection (36.67%) were the highest category displaying severe discrimination. Severe discrimination was highest among respondents with a middle school education level (39.01%), and lowest among respondent with a high school education level (31.23%). By occupation, the discrimination level of retirees (53.49%) was the highest, followed by professional and technical staff (42.67%) and the unemployed (41.75%), with physicians (12.9%) and students (20.97%) displaying significantly lower discrimination levels (*p* < 0.001). Severe discrimination in the central region (42.13%) was significantly higher than the eastern (37.03%) and western region (25.98%). Participants who had paid for vaccines for their family members or themselves in the last year (26.44%) showed a lower severe discrimination level than participants who never paid for a vaccination (32.78%, *p* < 0.001) or paid for a vaccine more than one year ago (37.94%, *p* < 0.001). The respondents who scored lower of COVID-19 knowledge displayed higher severe discrimination (40.66%) than those with a high COVID-19 knowledge score (26.96%, *p* < 0.001).

### 3.3. Discrimination against COVID-19 Recovered Patients between Unvaccinated and Intended Vaccinated Respondents

As shown in Table 3, the median discrimination score against recovered COVID-19 patients of unvaccinated respondents (6) was significantly higher than intended vaccinated respondents (2) (*p* < 0.001). The percentage of respondents displaying no discrimination towards recovered COVID-19 patients increased from 20.24% to 37.74%, and those with a discrimination score of 12 fell from 14.09% to 9.93%, after COVID-19 vaccination.

Figure 1 illustrates participants’ attitudes toward recovered COVID-19 patients through six life events. For unvaccinated respondents, just half, 51.45% were willing to work together and 50.65% were willing to accept gifts from recovered COVID-19 patients, but only 36.56% were willing to have dinner, 41.31% to hug, 29.74% to have children play and 29.45% to have children marry recovered COVID-19 patients. For intended vaccinated respondents, the percentage willing to participate with recovered COVID-19 patients increased significantly (*p* < 0.001) with 66.55% willing to accept gifts, 65.25% to work, 58.9% to have dinner, 61.93% to hug; 51.75% to have children play and 43.25% to have children marry recovered COVID-19 patients. Although the intended vaccinated respondents show a significantly lower level of discrimination, between one third and two fifths of intended vaccinated respondents displayed discriminatory behavior towards COVID-19 recovered patients.

### 3.4. Results of Multiple Ordered Logistic Regression

Built with independent variables which had statistical significance in Pearson chi-square test in Table 2, a multivariate ordered logistic regression model in Table 4 analyzed the relationship between the COVID-19 related discrimination level and the independent variables for intended vaccinated respondents. Sex, age, education level, occupation, region, awareness of vaccine effectiveness and infection risk, and knowledge score were significant determinants of the discrimination level against COVID-19 for respondents who intended to be vaccinated (*p* < 0.05). Experience of paying for vaccines had no influence on the COVID-19 associated discrimination.

The female discrimination level was higher than the male level (OR = 1.36, *p* < 0.001) and the discrimination level of all age groups were higher than the 18–27 years old age group, with respondents aged over 58 years old displaying the highest discrimination level (OR = 2.899, *p* < 0.001). The discrimination level of respondents who had a high school (OR = 1.63, *p* = 0.007) or above high school education level (OR = 1.602, *p* = 0.006) was higher than that of respondents who had a primary and below education level. The discrimination level of all occupations was higher than physicians, with retirees revealing the highest discrimination level (OR = 4.189), followed by migrant workers (OR = 4.035, *p* < 0.05). The respondents from the central region had the highest discrimination level (OR = 1.223, *p* = 0.05), followed by the respondents from eastern area (OR = XX *p* = 0.05), with the lowest discrimination in the western region (OR = 0.705, *p* < 0.001). The respondents who rated the COVID-19 vaccine effectiveness as “high” had significantly lower discrimination levels than respondents who rated the vaccine efficacy as “low” (OR = 0.155, *p* = 0.002). The participants who perceived that the COVID-19 infection risk was high, had significantly higher discrimination than people who thought the COVID-19 infection risk was low (OR = 1.35, *p* = 0.002). The discrimination level of high COVID-19 knowledge score group was lower than that of low COVID-19 knowledge score group (OR = 0.608, *p* < 0.001).

## 4. Discussion

Vaccination decreased the percentage of respondents who had COVID-19 related discrimination from 79.76% to 62.26%, with a reduction rate of 21.94%. Vaccination also decreased the percentage of respondents who answered “no” or “it depends” for all six life events. For both unvaccinated and intended vaccinated respondents, the percent of medium and severe discrimination of the life event “parents should allow their children to marry recovered COVID-19 patients” was the highest. The second highest discriminatory life event was “parents should let their children play with recovered COVID-19 patients”, followed by having dinner and shake handing or hugging recovered COVID-19 patients. The discrimination increased as the intimacy of the life events rose.

Perhaps the most surprising finding was that COVID-19 discrimination still existed at a high level after intended COVID-19 vaccination, with only 37.74% of the participants without any prejudice and 34.11% displaying severe discrimination. Sex, age, education level, occupation, region, awareness of vaccine effectiveness and infection risk, and knowledge score was significantly associated with COVID-19 related discrimination among intended vaccinated respondents. Experience of paying for vaccines was not significant associated with COVID-19 discrimination. We recommend that interventions should be taken to reduce COVID-19 discrimination after COVID-19 immunization, not only information campaigns directed towards the pre-inoculation population.

For intended vaccinated respondents, females displayed higher COVID-19 discrimination than males, which is consistent with a past HBV research where women had higher discrimination levels towards HBV-infected patients and carriers than men in rural China [23]. Discrimination by intended vaccinated respondents against recovered COVID-19 patients increased with age, with respondents over 58 years old displaying the highest discrimination level. A previous study also found that older healthy adults (46–55 years old) showed more SAD than younger adults (16–25 years old) towards AIDS patients [21]. Respondents with high school and above high school education level displayed higher discrimination level than respondents with primary and below education level. This finding mirrors a previous study showing that individuals with higher education level shown higher levels of discrimination against HIV-infected people [34], but is inconsistent with a study in rural China, which showed that people with higher education level tended to have less discrimination against HBV-infected patients [23]. Occupation influenced the intended vaccinated respondents’ discrimination level against recovered COVID-19 patients. Physicians exhibited the lowest discrimination level, while the retirees displayed the highest discrimination level, followed by migrant workers. Leng et al. [22] found unemployed respondents displayed higher HBV associated discrimination, but not by other researchers [18,21,23].

There also were geographical differences in COVID-19 discrimination. Respondents from central China showed the most severe discrimination, followed by the eastern region, with discrimination lowest in the western region. The central region was the area where COVID-19 originated in China and where 66.21% of reported cases (68168/102958) and 93.69% of death (4545/4851) occurred [3]. The western region had more minority people, so ethnicity, as well as geographical differences, should be studied in future research.

Given the high discrimination levels by intended vaccinated respondents against recovered COVID-19 patients, there is a need to inform the population about the causes and risks of COVID-19 and the effectiveness of the COVID-19 vaccine. Our study makes two specific contributions. First, our study identified post-COVID-19 vaccinated target groups for tailored intervention strategies, comprising females, the older aged, people with high school and above education level, retirees and migrant workers, and residents in central China. Second, our finding that intended vaccinated respondents’ cognition of COVID-19 vaccine effectiveness had the greatest effect on lowering discrimination against recovered COVID-19 patients suggests COVID-19 information campaigns should particularly focus on the vaccine’s post-inoculation efficacy.

General COVID-19 information campaigns should continue. Respondents who believe they were in a high risk of COVID-19 infection displayed high discrimination against recovered COVID-19 patients. Past research also found that the fear of HBV infection was a predominant factor in discrimination against HBV patients or carriers [23]. General information campaigns should provide accurate information on the risk of COVID-19 infection both before and after inoculation of the COVID-19 vaccine. Such information campaigns increase public trust in COVID-19 vaccines and their effectiveness and decrease the intended vaccinated fear of recovered COVID-19 patients. Since the discrimination level of respondents with a high COVID-19 knowledge score was lower than those with low COVID-19 knowledge score, our results support further information campaigns. This is also consistent with previous research on discrimination against HIV-, HBV- and HCV-infected patients that found the lack of knowledge was the main cause of discrimination [22,23,31]. Finally, targeted and general COVID-19 knowledge campaigns on transmissions, preventions and the vaccines will counter the spread of COVID-19 misinformation, provide accurate information about COVID-19 and reduce discrimination against recovered COVID-19 patients [14,16,35].

### Strengths and Limitations

This study has two main strengths. First, this is the first study to assess the discrimination against recovered COVID-19 patients by the intended COVID-19 vaccinated in China. Second, the data are from nationwide study, covering both rural and urban areas and three (eastern, central, and western) regions in China.

There were four major limitations. First, all respondents were not vaccinated, but asked “if you have been vaccinated against COVID-19” what their response would be. Our data were collected before widespread vaccinations and future research should be conducted among people who had received the COVID-19 vaccine. Our results still provide a good assessment as vaccination behavior was highly associated with vaccination willingness [36,37]. Second, only six life events were used to assess participants’ COVID-19 discrimination in our questionnaire. Although these six life events were consistent with similar studies [22,23], a more complex COVID-19 SAD scale should be developed in further studies. Third, the sample was limited to only 100 respondents in each city across three regions in China. Future studies should consider larger sample sizes. Last, the information on the COVID-19 vaccination status of respondents’ family members was not collected, which might influence the results of measured questions about family members.

## 5. Conclusions

Only 37.74% of the intended vaccinated respondents displayed no discrimination against recovered COVID-19 patients, which indicates high levels of discrimination against recovered COVID-19 patients in China. Discrimination against COVID-19 non-recovered suffers was likely to be higher [38]. The COVID-19 vaccination did reduce COVID-19 discrimination against recovered COVID-19 patients by 21.94%. There were significant correlations between sex, age, education level, occupation, region, respondents’ awareness of vaccine effectiveness and infection risk, knowledge score, and the discrimination against recovered COVID-19 patients. Females, the older aged, people with high school and above education level, the retirees and migrant workers, and residents in central China were identified as key intervention targets to reduce discrimination against recovered COVID-19 patients when people have been vaccinated. We recommend targeted publicity and education campaigns about the efficacy of COVID-19 vaccines for the COVID-19 vaccinated, as well as general information COVID-19 campaigns to decrease the fear of being infected by COVID-19, increase the knowledge of COVID transmission and prevention and to confirm vaccine effectiveness.

## Figures and Tables

**Figure 1 vaccines-09-00490-f001:**
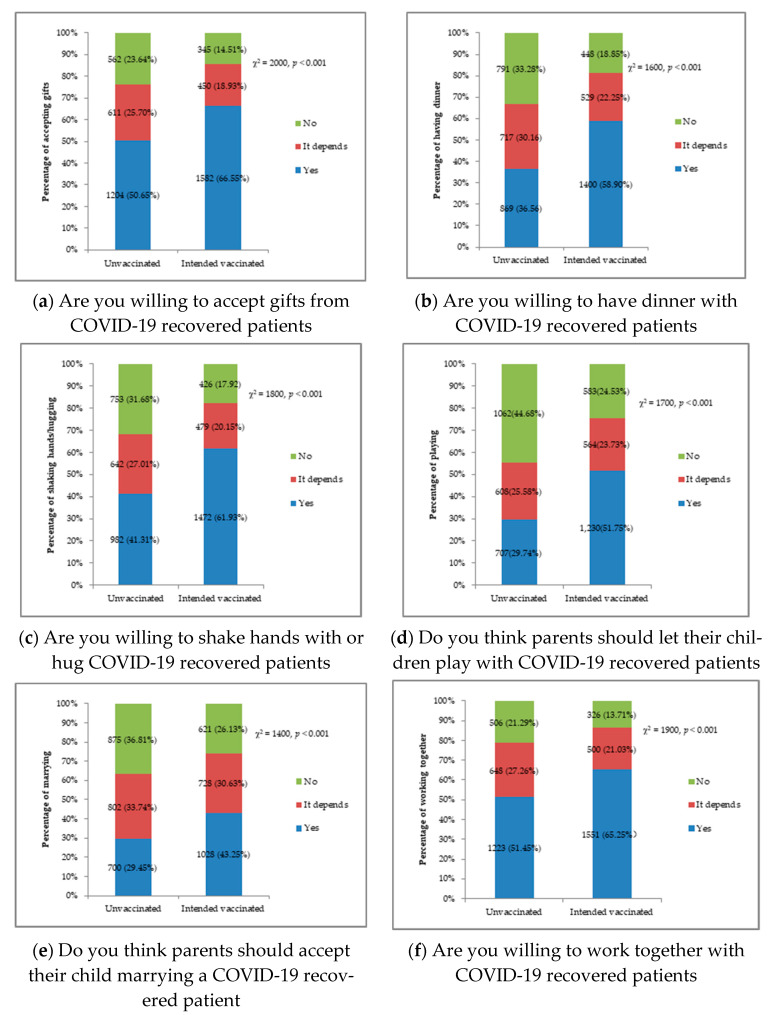
Discrimination level between unvaccinated and intended vaccinated respondents by each life event.

**Table 1 vaccines-09-00490-t001:** Life events measures of discrimination against recovered COVID-19 patients.

Vaccination	Measurement Events
Unvaccinated	1. Are you willing to accept gifts from COVID-19 recovered patients?
	2. Are you willing to have dinner with COVID-19 recovered patients?
	3. Are you willing to shake hands with or hug COVID-19 recovered patients?
	4. Do you think parents should let their children play with COVID-19 recovered patients?
	5. Do you think parents should accept their child marrying a COVID-19 recovered patient?
	6. Are you willing to work together with COVID-19 recovered patients?
Intended vaccinated	1. Are you willing to accept gifts from COVID-19 recovered patients if you have been vaccinated against COVID-19?
	2. Are you willing to have dinner with COVID-19 recovered patients if you have been vaccinated against COVID-19?
	3. Are you willing to shake hands with or hug COVID-19 recovered patients if you have been vaccinated against COVID-19?
	4. Do you think parents should let their children play with COVID-19 recovered patients if their children have been vaccinated against COVID-19?
	5. Do you think parents should accept their child marrying a COVID-19 recovered patient if their children have been vaccinated against COVID-19?
	6. Are you willing to work together with COVID-19 recovered patients if you have been vaccinated against COVID-19?

**Table 2 vaccines-09-00490-t002:** Discrimination levels and characteristics of respondents (N (%)).

Variables		Total	Mild/No	Medium	Severe	χ^2^	*p*
Sex	Male	1154(48.55%)	480(41.59%)	302(26.17%)	372(32.24%)	14.28	0.001
	Female	1223(51.45%)	417(34.10%)	367(30.01%)	439(35.90%)		
Age	18–27	915(38.49%)	449(49.07%)	245(26.78%)	221(24.15%)	107.16	<0.001
	28–37	340(14.30%)	121(35.59%)	94(27.65%)	125(36.76%)		
	38–47	460(19.35%)	137(29.78%)	143(31.09%)	180(39.13%)		
	48–57	426(17.92%)	131(30.75%)	125(29.34%)	170(39.91%)		
	58+	236(9.93%)	59(25.00%)	62(26.27%)	115(48.73%)		
Income	Lowest	456(19.18%)	186(40.79%)	128(28.07%)	142(31.14%)	3.50	0.899
	Lower	350(14.72%)	132(37.71%)	97(27.71%)	121(34.57%)		
	Medium	514(21.62%)	186(36.19%)	148(28.79%)	180(35.02%)		
	Higher	421(17.71%)	153(36.34%)	122(28.98%)	146(34.68%)		
	Highest	636(26.76%)	240(37.74%)	174(27.36%)	222(34.91%)		
Education	Primary school and below	272(11.44%)	92(33.82%)	83(30.51%)	97(35.66%)	16.89	0.010
	Middle school	405(17.04%)	124(30.62%)	123(30.37%)	158(39.01%)		
	High school	349(14.68%)	134(38.40%)	106(30.37%)	109(31.23%)		
	Above high school	1351(56.84%)	547(40.49%)	357(26.42%)	447(33.09%)		
Occupation	Farmer	31(1.30%)	17(54.84%)	10(32.26%)	4(12.90%)	128.70	<0.001
	Migrant worker	278(11.70%)	99(35.61%)	69(24.82%)	110(39.57%)		
	Enterprise staff	289(12.16%)	88(30.45%)	91(31.49%)	110(38.06%)		
	Individual industrialist	323(13.59%)	108(33.44%)	88(27.24%)	127(39.32%)		
	Teacher	221(9.30%)	67(30.32%)	68(30.77%)	86(38.91%)		
	Physician	159(6.69%)	60(37.74%)	49(30.82%)	50(31.45%)		
	Civil servant	120(5.05%)	34(28.33%)	38(31.67%)	48(40.00%)		
	Professional and technical staff	75(3.16%)	27(36.00%)	16(21.33%)	32(42.67%)		
	Unemployed	103(4.33%)	32(31.07%)	28(27.18%)	43(41.75%)		
	Student	639(26.88%)	335(52.43%)	170(26.60%)	134(20.97%)		
	Retiree	86(3.62%)	18(20.93%)	22(25.58%)	46(53.49%)		
	Others	53(2.23%)	12(22.64%)	20(37.74%)	21(39.62%)		
Medical insurance	Yes	2289(96.30%)	866(37.83%)	642(28.05%)	781(34.12%)	0.36	0.835
	No	88(3.70%)	31(35.23%)	27(30.68%)	30(34.09%)		
Residence	Urban	1462(61.51%)	561(38.37%)	406(27.77%)	495(33.86%)	0.67	0.71
	Rural	915(38.49%)	336(36.72%)	263(28.74%)	316(34.54%)		
Self-rated health	Bad	73(3.07%)	24(32.88%)	19(26.03%)	30(41.10%)	6.2	0.181
	Medium	564(23.73%)	193(34.22%)	164(29.08%)	207(36.70%)		
	Good	1740(73.20%)	680(39.08%)	486(27.93%)	574(32.99%)		
Region	Eastern	748(31.47%)	282(37.70%)	189(25.27%)	277(37.03%)	53.97	<0.001
	Central	686(28.86%)	211(30.76%)	186(27.11%)	289(42.13%)		
	Western	943(39.67%)	404(42.84%)	294(31.18%)	245(25.98%)		
Vaccine effectiveness	Low	17(0.72%)	2(11.76%)	2(11.76%)	13(76.47%)	105.94	<0.001
	Medium	301(12.66%)	57(18.94%)	71(23.59%)	173(57.48%)		
	High	2059(86.62%)	838(40.70%)	596(28.95%)	625(30.35%)		
Risk of infection	Low	1107(46.57%)	447(40.38%)	312(28.18%)	348(31.44%)	14.49	0.006
	Medium	670(28.19%)	256(38.21%)	171(25.52%)	243(36.27%)		
	High	600(25.24%)	194(32.33%)	186(31.00%)	220(36.67%)		
Paid for vaccine	No	1028(43.25%)	433(42.12%)	258(25.10%)	337(32.78%)	32.56	<0.001
	Within last year	329(13.84%)	138(41.95%)	104(31.61%)	87(26.44%)		
	More than one year ago	1020(42.91%)	326(31.96%)	307(30.10%)	387(37.94%)		
Infections of relatives and friends	No	2310(97.18%)	870(37.66%)	651(28.18%)	789(34.16%)	5.16	0.271
	Yes	36(1.51%)	15(41.67%)	13(36.11%)	8(22.22%)		
	Not sure	31(1.30%)	12(38.71%)	5(16.13%)	14(45.16%)		
Knowledge score	Low group	1242(52.25%)	399(32.13%)	338(27.21%)	505(40.66%)	55.12	<0.001
	High group	1135(47.75%)	498(43.88%)	331(29.16%)	306(26.96%)		

**Table 3 vaccines-09-00490-t003:** Discrimination scores between unvaccinated and intended vaccinated respondents.

Discrimination Scores	Unvaccinated		Intended Vaccinated		χ^2^	*p*
	N	%	N	%		
0	481	20.24	897	37.74	283.92	*p* < 0.001
1	108	4.54	171	7.19		
2	171	7.19	204	8.58		
3	103	4.33	114	4.80		
4	131	5.51	99	4.16		
5	122	5.13	81	3.41		
6	361	15.19	298	12.54		
7	124	5.22	73	3.07		
8	133	5.60	74	3.11		
9	82	3.45	36	1.51		
10	109	4.59	55	2.31		
11	117	4.92	39	1.64		
12	335	14.09	236	9.93		

**Table 4 vaccines-09-00490-t004:** Multivariate ordered logistic regression.

Variables		β	S. E.	z	*p*	OR (95%CI)
Sex	Male	(Reference group)				
	Female	0.31	0.08	3.86	<0.001	1.360(1.163,1.590)
Age	18–27	(Reference group)				
	28–37	0.33	0.16	2.14	0.032	1.395(1.029,1.892)
	38–47	0.58	0.15	3.81	<0.001	1.791(1.327,2.417)
	48–57	0.54	0.16	3.41	0.001	1.719(1.259,2.349)
	58+	1.06	0.20	5.19	<0.001	2.899(1.940,4.332)
Education	Primary school and below	(Reference group)				
	Middle school	0.22	0.16	1.42	0.155	1.250(0.919,1.699)
	High school	0.49	0.18	2.68	0.007	1.630(1.141,2.329)
	Above high school	0.47	0.17	2.72	0.006	1.602(1.141,2.249)
Occupation	Physician	(Reference group)				
	Farmer	1.22	0.38	3.17	0.002	3.386(1.594,7.192)
	Migrant worker	1.39	0.38	3.71	<0.001	4.035(1.929,8.439)
	Employee	1.34	0.37	3.66	<0.001	3.834(1.867,7.873)
	Self-employed	1.34	0.38	3.56	<0.001	3.819(1.826,7.987)
	Teacher	0.80	0.38	2.12	0.034	2.229(1.061,4.682)
	Civil servant	1.32	0.39	3.40	0.001	3.757(1.753,8.053)
	Professional and technical staff	1.34	0.42	3.22	0.001	3.812(1.687,8.617)
	Unemployed	1.21	0.41	2.93	0.003	3.358(1.495,7.539)
	Student	0.85	0.38	2.25	0.024	2.345(1.118,4.922)
	Retiree	1.43	0.42	3.40	0.001	4.189(1.836,9.560)
	Others	1.35	0.44	3.09	0.002	3.857(1.637,9.087)
Region	Eastern	(Reference group)				
	Central	0.20	0.10	1.96	0.050	1.223(1.000,1.497)
	Western	−0.35	0.10	−3.66	<0.001	0.705(0.585,0.850)
Vaccine effectiveness	Low	(Reference group)				
	Medium	−0.81	0.60	−1.35	0.178	0.445(0.137,1.445)
	High	−1.87	0.59	−3.16	0.002	0.155(0.049,0.493)
Risk of infection	Low	(Reference group)				
	Medium	0.14	0.09	1.49	0.136	1.152(0.957,1.386)
	High	0.30	0.10	3.10	0.002	1.350(1.117,1.632)
Paid for vaccine	No	(Reference group)				
	Within last year	−0.23	0.13	−1.79	0.074	0.797(0.622,1.022)
	More than one year ago	0.02	0.09	0.19	0.851	1.018(0.847,1.222)
COVID-19 knowledge score	Low group	(Reference group)				
	High group	−0.50	0.08	−6.23	<0.001	0.608(0.520,0.711)

## Data Availability

The data presented in this study are available on request from the corresponding author. The data are not publicly available due to multi-cooperation with Wuhan University, Shandong University and Nanjing Medical University. The corresponding author will facilitate a discussion with these three universities for data access on a reasonable request.

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
