# Peer review of "Will COVID-19 Vaccinations End Discrimination against COVID-19 Patients in China? New Evidence on Recovered COVID-19 Patients"

_vaccines, 2021, doi:10.3390/vaccines9050490_

Round 1
Reviewer 1 Report
Dear editor, Dear authors,
I have read with an interest this paper. The authors took up a very current and important topic and assessed the discrimination against recovered COVID-19 patients by the COVID-19 vaccinated, which is the first in the country. The following issues need correction:
The introduction should be more concise. Moreover, it should be concluded by the hypothesis of the study!
Sections 3.1 If you write, that 49% are female there is no point in writing that 51% is male.... I would suggest to shorten this section a little bit.
Table 2. I would suggest to put the percentages in parentheses to make the table more readable.
What were the inclusion/exclusions criteria ?
One of the most interesting things about vaccination is its link with the willingness to get vaccinated. In my opinion, it should be discussed. Please find this paper published in MDPI journals as helpful:
- https://www.mdpi.com/2076-393X/9/3/218
- https://www.mdpi.com/2076-393X/9/2/128
Also it would be good to mention about general well-being during pandemic. There are few studies in this subject published in MDPI journals, e.g.: https://www.mdpi.com/1660-4601/17/20/7417
I am looking forward to a better version! Good luck!
Author Response
Dear editor, Dear authors,
I have read with an interest this paper. The authors took up a very current and important topic and assessed the discrimination against recovered COVID-19 patients by the COVID-19 vaccinated, which is the first in the country. The following issues need correction:
The introduction should be more concise. Moreover, it should be concluded by the hypothesis of the study!
Authors’ response:
The introduction was revised and the hypothesis of study was highlighted in lines 128-130.
Sections 3.1 If you write, that 49% are female there is no point in writing that 51% is male.... I would suggest to shorten this section a little bit.
Authors’ response:
The Section 3.1 was shortened. (lines 207-219)
Table 2. I would suggest to put the percentages in parentheses to make the table more readable.
Authors’ response:
Table 2 was revised following reviewer’s comment.
What were the inclusion/exclusions criteria ?
Authors’ response:
The selection methods of respondents were explained in Section ‘2.1. Data Source and Sample’. (lines 135-153)
One of the most interesting things about vaccination is its link with the willingness to get vaccinated. In my opinion, it should be discussed. Please find this paper published in MDPI journals as helpful:
- https://www.mdpi.com/2076-393X/9/3/218
- https://www.mdpi.com/2076-393X/9/2/128
Also it would be good to mention about general well-being during pandemic. There are few studies in this subject published in MDPI journals, e.g.: https://www.mdpi.com/1660-4601/17/20/7417
I am looking forward to a better version! Good luck!
Authors’ response:
Following reviewer’s suggestions, these discussions and references were added in lines 47-48 and 382-384 and references [1][37][38].
Reviewer 2 Report
Thank you for inviting me to peer review this very interesting study by Li and colleagues, who interviewed almost 2,700 people from China to explore whether the discrimination against people recovered from COVID-19 will persist after vaccinations. The population was geographically and economically balanced.
As described by the authors, the main limitation of this study is that the "vaccinated" population was in reality unvaccinated people who were asked how would they feel if they were vaccinated. However, the perceptions may change after broad vaccination and after experiencing the lifechanges and sense of safety that will be associated with vaccination. Moreover, this population is called "vaccinated" throughout the manuscript and only in the 2nd last paragraph it is revealed that they are not in fact vaccinated but this is an imaginary scenario. this needs to be clarified throughout the manuscript and the group should not be called "vaccinated", as this is a misleading term.
Moreover, it would be interesting to present (in the main article or online appendix) the detailed responses of the participants to each of the scenarios described, as scenarios evaluated different aspects of daily life.
Another limitation of the study is that the investigators asked the "vaccinated" group about their willingness to interact with COVID-19 recovered patients "if the respondents had been vaccinated against COVID-19". However, it was not clarified whether their family members were also vaccinated in this "imaginary" scenario. Therefore, people might declined these interactions out of concern for their families. This needs to be discussed.
Finally, the authors should describe other similar studies that may have been published in the discussion section.
Author Response
Thank you for inviting me to peer review this very interesting study by Li and colleagues, who interviewed almost 2,700 people from China to explore whether the discrimination against people recovered from COVID-19 will persist after vaccinations. The population was geographically and economically balanced.
As described by the authors, the main limitation of this study is that the "vaccinated" population was in reality unvaccinated people who were asked how would they feel if they were vaccinated. However, the perceptions may change after broad vaccination and after experiencing the lifechanges and sense of safety that will be associated with vaccination. Moreover, this population is called "vaccinated" throughout the manuscript and only in the 2nd last paragraph it is revealed that they are not in fact vaccinated but this is an imaginary scenario. this needs to be clarified throughout the manuscript and the group should not be called "vaccinated", as this is a misleading term.
Authors’ response:
Thanks for reviewer’s valuable suggestion. The term “vaccinated” was replaced by “intended vaccinated” throughout the manuscript, with a discussion on lines 303-373 and in the Limitation section (lines 379-384).
Moreover, it would be interesting to present (in the main article or online appendix) the detailed responses of the participants to each of the scenarios described, as scenarios evaluated different aspects of daily life.
Authors’ response:
The responses of the participants to each of the scenarios described are explained in lines 256-272 and Figure 1.
Another limitation of the study is that the investigators asked the "vaccinated" group about their willingness to interact with COVID-19 recovered patients "if the respondents had been vaccinated against COVID-19". However, it was not clarified whether their family members were also vaccinated in this "imaginary" scenario. Therefore, people might declined these interactions out of concern for their families. This needs to be discussed.
Authors’ response:
This limitation was added in Section “Strengths and Limitations”. (lines 389-391)
Finally, the authors should describe other similar studies that may have been published in the discussion section.
Authors’ response:
The similar studies were described in Section “Introduction” (lines 73-120) and “Discussion” (lines 323-373).
Round 2
Reviewer 1 Report
All changes have been made, the manuscript is now improved and ready to be published.
Good work !
Reviewer 2 Report
The authors have now addressed the peer review comments